# Ethograms predict visual fear conditioning status in rats

David C Williams, Amanda Chu, Nicholas T Gordon, Aleah M DuBois, Suhui Qian, Genevieve Valvo, Selena Shen, Jacob B Boyce, Anaise C Fitzpatrick, Mahsa Moaddab, Emma L Russell, Liliuokalani H Counsman, Michael A McDannald*

Boston College Department of Psychology and Neuroscience, Chestnut Hill, United States

**Abstract** Recognizing and responding to threat cues is essential to survival. Freezing is a predominant threat behavior in rats. We have recently shown that a threat cue can organize diverse behaviors beyond freezing, including locomotion (Chu et al., 2024). However, that experimental design was complex, required many sessions, and had rats receive many foot shock presentations. Moreover, the findings were descriptive. Here, we gave female and male Long Evans rats cue light illumination paired or unpaired with foot shock (eight total) in a conditioned suppression setting using a range of shock intensities (0.15, 0.25, 0.35, or 0.50 mA). We found that conditioned suppression was only observed at higher foot shock intensities (0.35 mA and 0.50 mA). We constructed comprehensive temporal ethograms by scoring 22,272 frames across 12 behavior categories in 200-ms intervals around cue light illumination. The 0.50 mA and 0.35 mA shock-paired visual cues suppressed reward seeking, rearing, and scaling, as well as light-directed rearing and light-directed scaling. These shock-paired visual cues further elicited locomotion and freezing. Linear discriminant analyses showed that ethogram data could accurately classify rats into paired and unpaired groups. Using complete ethogram data produced superior classification compared to behavior subsets, including an immobility subset featuring freezing. The results demonstrate diverse threat behaviors – in a short and simple procedure – containing sufficient information to distinguish the visual fear conditioning status of individual rats.

*For correspondence:
michael.mcdannald@bc.edu

## Editor's evaluation

This report provides a useful characterization of the behaviors evoked by a Pavlovian conditioned stimulus (CS) paired with foot shock in male and female rats. The aim of the study was to assess the generalizability of the authors' past findings (Chu et al., 2024), established using an auditory CS (a tone), to a visual CS (a light). The reviewers appreciated the extensive nature of the task undertaken: there was agreement that the methods and analyses used to produce the study findings are solid. This work will be of interest to those who study associative learning, fear, ethological assessment, and behavior more broadly.

## Introduction

In a typical Pavlovian fear conditioning procedure, a neutral cue is paired with foot shock. As a result, the shock-paired cue acquires many properties (*McDannald, 2023*). The most well-known property is that a shock-paired cue (or context) can elicit freezing (*Bolles and Collier, 1976*). Pavlovian conditioning procedures measuring freezing have been used to great effect to reveal neural bases of fear

(*LeDoux et al., 1988*; *De Oca et al., 1998*; *Goosens and Maren, 2001*; *Koo et al., 2004*; *Monfils et al., 2009*; *Li et al., 2013*; *Massi et al., 2023*).

A shock-paired cue can also suppress reward seeking (*Estes and Skinner, 1941*), a phenomenon referred to as conditioned suppression (*Killcross et al., 1997*). Using conditioned suppression procedures, we recently demonstrated that a shock-paired auditory cue elicits locomotion, rearing, and jumping, in addition to freezing (*Chu et al., 2024*). Though consistent with some studies (*Gruene et al., 2015*; *Totty et al., 2021*; *Mitchell et al., 2022*; *Le et al., 2024*; *Borkar et al., 2024*; *Chanthongdee et al., 2024*), observing such a diversity of fear-conditioned behaviors is uncommon. In rats, and across a variety of settings, freezing is a predominant fear-conditioned behavior (*Lay et al., 2020*; *Liu et al., 2022*; *Trott et al., 2022*; *Williams-Spooner et al., 2022*; *Fam et al., 2023*; *Ng and Sangha, 2023*; *Ng et al., 2024*; *Yau et al., 2024*). It is possible the diversity we observed was due to the complexity of our experimental design (2–3 cues, including an uncertainty cue), the specific stimuli used (complex auditory cues), or the numerous shock presentations (50+). Further, our results were descriptive. If there is a robust relationship between a rat's fear conditioning status (paired vs. unpaired cue and shock presentations) and these diverse behaviors, then we should be able to predict an individual's status from ethograms spanning all behaviors.

The goal of the present experiments was to comprehensively quantify behaviors evoked by a shock-paired cue in a simple, visual fear conditioning setting. We then sought to determine if ethograms can predict rats' fear conditioning status (paired vs. unpaired). Mildly food-deprived, Long Evans rats (half female) were trained to nose poke for food, then received Pavlovian fear conditioning over a baseline of rewarded poking. All rats received a 10-s visual cue paired or unpaired with foot shock. First, we identified the lowest shock intensity that would support conditioned suppression (in the manner of *Holland, 1979*). In Experiment 1, rats received 0.15 or 0.25 mA foot shock. In Experiment 2, rats received 0.35 or 0.50 mA foot shock. All rats received eight total shock presentations, thereby making these experiments simpler and shorter than our prior work (*Chu et al., 2024*). Extinction testing (four total cue light illuminations) was designed to reveal behaviors acquired by the visual cue during conditioning, rather than the reduction of those behaviors over trials and sessions.

TTL-triggered cameras were programmed to capture frames at 200-ms intervals before, during, and following cue light illumination. For Experiment 2, 22,272 frames were hand scored for 12 behaviors under categories of Immobility (freezing and stretching), Horizontal movement (locomotion and backpedaling), Vertical movement (rearing, scaling, and jumping), and Reward-related behavior (nose port and food cup). Light-directed variants of each vertical movement type were differentiated. Ethograms spanning all 12 behaviors were constructed from the scored frames. Multivariate and univariate ANOVA performed on the ethogram data revealed behaviors that differed between groups. Linear discriminant analyses determined if group membership (paired vs. unpaired) can be classified from ethogram data.

## Results

Our first objective was to determine the lowest foot shock intensity capable of supporting conditioned suppression to a visual cue. Rats were shaped to nose poke in a central port to receive a food reward, delivered via a cup directly below. The cue light was positioned above the central port. Cue light illumination resulted in bright, focal illumination of the light itself as well as diffuse illumination of the entire context, which was otherwise dark (*Figure 1A*). In Experiment 1, rats received cue light illumination paired or unpaired with a 0.25 or 0.15 mA foot shock. In Experiment 2, rats received cue light illumination paired or unpaired with a 0.50 or 0.35 mA foot shock (*Figure 1B*).

### Higher, but not lower, shock intensities support conditioned suppression to a visual cue

Neither the 0.15 mA nor the 0.25 mA foot shock intensity supported conditioned suppression to the visual cue. Paired and unpaired rats in both shock intensity conditions showed identical, decreasing suppression ratio patterns across all 20 trials (*Figure 1C*). In support, ANOVA for suppression ratio [within factors: session (1–5; two pre-exposure, two conditioning, and one extinction), and trial (1–4); between factors: sex (female vs. male), group (paired vs. unpaired), and intensity (0.25 mA vs. 0.15 mA); *Figure 1C*] found no significant main effect of group ($F_{1,24} = 3.67$, p=0.067), and no significant

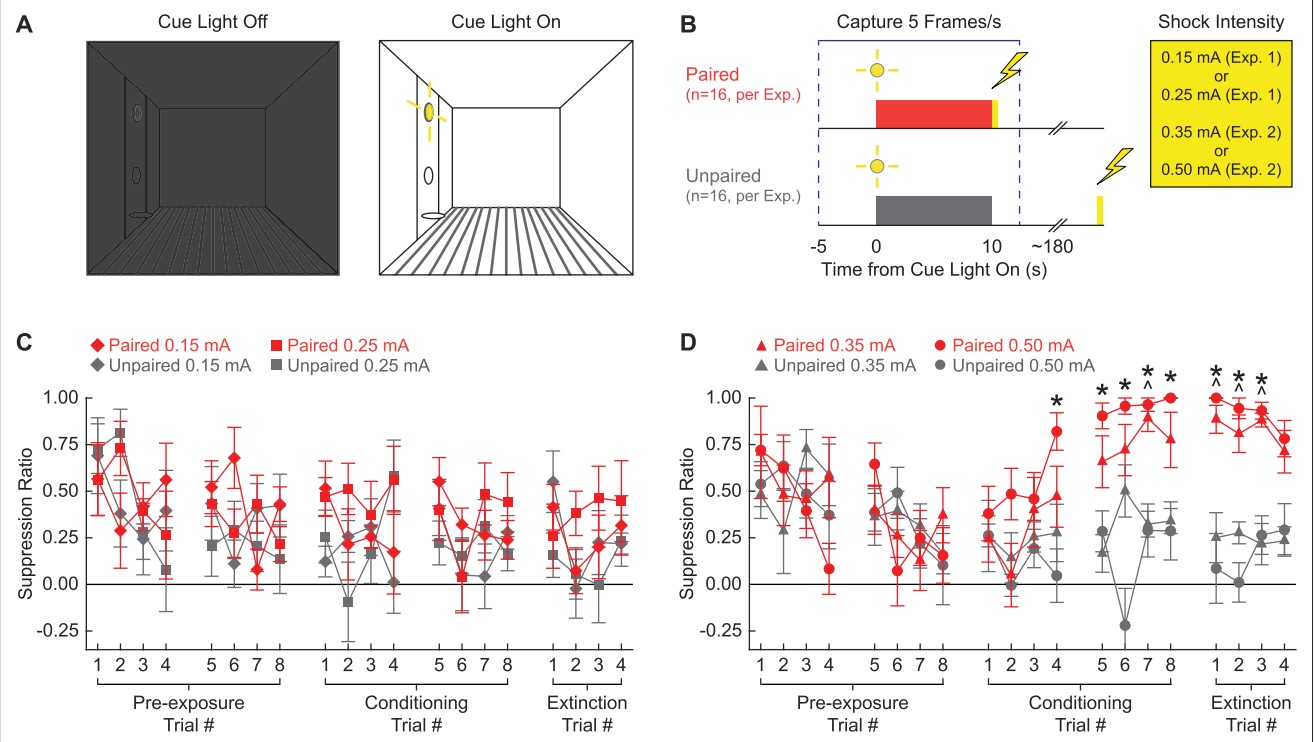

**Figure 1.** Experimental design and suppression ratio results. (**A**) Behavioral testing occurred in a dark, Med Associates chamber equipped with a food cup, port, and cue light on one wall (left). Cue light illumination provided a discrete visual cue in addition to generally illuminating the chamber. (**B**) Rats in each experiment were divided into paired (red; n = 16) and unpaired (gray; n = 16) conditions. Experiment 1 used foot shock intensities of 0.15 mA and 0.25 mA, while Experiment 2 used foot shock intensities of 0.35 mA and 0.50 mA. Behavior frames were captured every 200 ms, 5 s prior to and 2.5 s following cue light illumination. (**C**) Mean ± SEM suppression ratio for paired (red) and unpaired (gray) rats receiving 0.15 mA (diamond) or 0.25 mA (square) foot shock is shown for the 20 cue light illuminations (pre-exposure trials 1–8, conditioning trials 1–8, and extinction trials 1–4). (**D**) Formatting, other than foot shock intensity (0.35 mA, triangles; 0.50 mA, circles), is identical to (**C**). ^0.35 mA, independent samples *t*-test p<0.0025. *0.50 mA, independent samples *t*-test p<0.0025.

The online version of this article includes the following figure supplement(s) for figure 1:

**Figure supplement 1.** Conditioned suppression across sessions in female and male rats (Experiment 2).

group interactions: group × session ($F_{4,96}$ = 0.61, p=0.66), and session × trial × group ($F_{12,288}$ = 0.77, p=0.68). Two-tailed, independent samples *t*-tests found no significant differences between the paired and unpaired rats receiving the 0.25 mA foot shock or the 0.15 mA foot shock for any of the 20 trials (all p>0.05). Further, ANOVA confined to the extinction session [within factors: trial (1–4); between factors: sex (female vs. male), group (paired vs. unpaired), and intensity (0.50 mA vs. 0.35 mA)] found no significant main effect of group ($F_{1,24}$ = 2.03, p=0.17).

By contrast, the 0.50 mA and the 0.35 mA foot shock intensities supported conditioned suppression to the visual cue. Paired and unpaired rats showed equivalent suppression ratios during pre-exposure, but diverged thereafter. Paired rats increased suppression ratios over the eight conditioning trials, with elevated suppression ratios apparent during the extinction session. Unpaired rats persisted in showing low suppression ratios during both conditioning and extinction trials. In support, ANOVA for suppression ratio [within factors: session (1–5; two pre-exposure, two conditioning, and one extinction), and trial (1–4); between factors: sex (female vs. male), group (paired vs. unpaired), and intensity (0.50 mA vs. 0.35 mA); *Figure 1D*] found a significant main effect of group ($F_{1,24}$ = 37.2, p=2.66 × $10^{-6}$), a significant group × session interaction ($F_{4,96}$ = 18.12, p=4.06 × $10^{-11}$), and significant group × session × trial interaction ($F_{12,288}$ = 2.28, p=0.009). ANOVA also showed a significant session × sex × group interaction ($F_{4,96}$ = 3.48, p=0.011). Female paired and unpaired rats diverged more greatly over the five sessions than the males (*Figure 1—figure supplement 1*). Conditioned suppression was apparent when extinction was analyzed alone. ANOVA for suppression ratios [within factors: trial (4); between factors: sex (female vs. male), group (paired vs. unpaired), and intensity (0.50 mA vs. 0.35

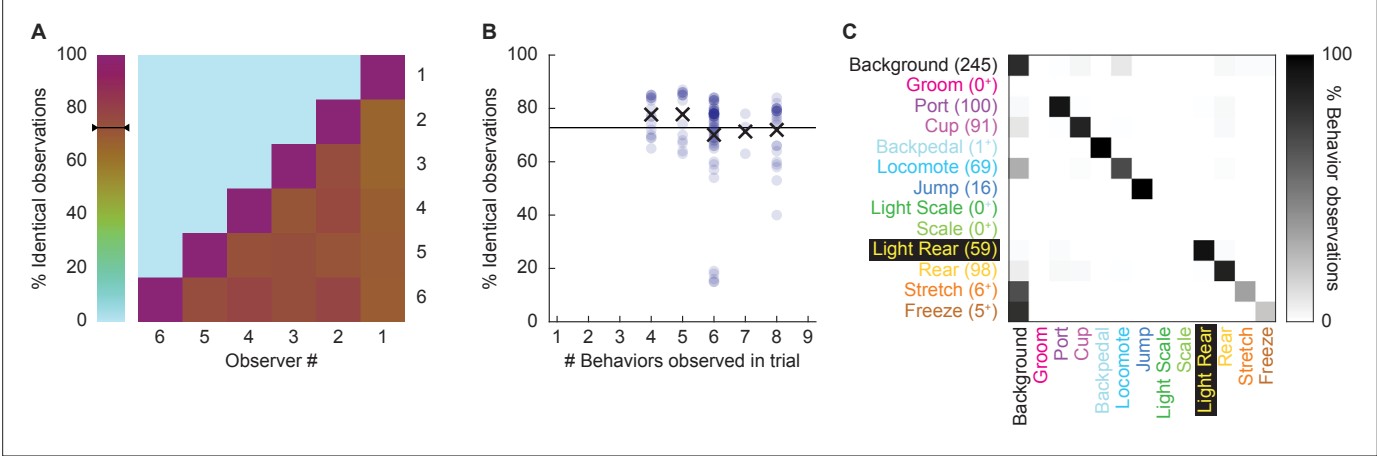

**Figure 2.** Hand scoring inter-rater reliability. (**A**) % identical observations for the comparison frames were calculated for each observer–observer pair. Brown to purple color indicates a higher % identical observations. The black bar on the scale shows the mean % identical observations across all observer–observer pairs. (**B**) % identical observations are broken down by the number of behaviors present during a trial, ranging from 4 to 8. Each point indicates a single observer–observer trial comparison. Large Xs indicate the mean % identical observations for all observer–observer comparisons for each number of behaviors. (**C**) A confusion matrix shows all observer–observer judgment pairs for the comparison trials. The y axis is observer 1 and x axis is observer 2. % behavior observations are plotted by row with black indicating 100%, white 0%, and shades of gray in between. The number of behavior observations is indicated for each in parentheses. Plus signs indicate behaviors for which fewer than 10 instances were observed.

mA); *Figure 1D*] revealed a significant main effect of group ($F_{1,24} = 154.01$, p=6.21 × 10$^{-12}$) and, as expected, a significant sex × group interaction ($F_{1,24} = 9.21$, p=0.006; *Figure 1—figure supplement 1*).

Independent samples *t*-tests comparing paired and unpaired conditioned suppression ratios on each of the 20 trials (p<0.0025; Bonferroni-corrected for 20 tests; significance indicated in *Figure 1D*) identified conditioning session 2 and extinction as the focus of differential nose poke suppression. For rats receiving the 0.35 mA foot shock, differential paired vs. unpaired nose poke suppression was observed on conditioning trial 7 and extinction trials 1–3. For rats receiving the 0.50 mA foot shock, differential nose poke suppression was observed on conditioning trials 4–8 and extinction trials 1–3.

## Scoring 12 behaviors with high inter-rater reliability and low inter-rater confusion

We defined and scored 12 behaviors in 200-ms intervals around cue light illumination. These behaviors were drawn from existing literature (*Holland, 1977*; *Fanselow, 1982*; *Chu et al., 2024*), as well as our observations. These behaviors were divided into four different categories: Immobile (freezing and stretching), Reward (cup and port), Vertical (rearing, scaling, and jumping), and Horizontal (locomotion and backpedaling). Within the Vertical category, we separated rearing and scaling that were specifically directed toward the light. This was done because light-directed behavior reflects stimulus orienting, while non-light-directed behavior reflects context exploration.

We focused our scoring efforts on Experiment 2: conditioning session 2 and the extinction session. During these sessions, paired, but not unpaired rats, receiving the 0.50 mA and 0.35 mA foot shock intensities demonstrated robust nose poke suppression. Six observers blind to group, sex, and intensity scored 22,272 total frames (see 'Materials and methods' for blinding details): 11,136 from the second conditioning session and 11,136 from the extinction session. Additionally, these six observers scored 696 frames spanning eight separate comparison trials. We determined inter-rater reliability by calculating the % identical observations for each observer-observer pair across the 696 frames (*Figure 2A*). The mean inter-rater reliability was 72.41%. Although inter-rater reliability was higher when fewer behaviors were present within a trial (e.g., 77.7% when three were present), the inter-rater reliability remained high, at 71.9%, even when eight behaviors were present (*Figure 2B*).

Disagreement rarely involved two observers assigning different, specific behaviors to a single frame. For example, a disagreement in which one observer assigns jump to a frame, while another observer assigns rear, is a seldom occurrence. Instead, disagreement occurred when one observer

assigned background to a frame while the other observer assigned a specific behavior to that frame. To quantify disagreements, we constructed a confusion matrix (*Figure 2C*) composed of every single-frame judgment for every observer–observer pair. Identical observations for every frame (zero confusion) appear as a diagonal line across the matrix (top left to bottom right). Confusing a specific behavior and background appears as a horizontal line on the top edge or a vertical line on the left edge. Confusing a specific behavior for another specific behavior appears as values occurring between the diagonal and edges. The confusion matrix shows the vast majority of values fall on the diagonal, with most of the remaining values on the left edge. The confusion matrix results reveal high scoring specificity.

## Ethograms reveal diverse fear-conditioned behaviors, including locomotion

We constructed 200-ms resolution ethograms for paired (*Figure 3A*) and unpaired (*Figure 3B*) rats from Experiment 2. For both groups, baseline behavior largely consisted of rearing, food cup, and port behavior, with very small amounts of freezing and locomotion. Paired and unpaired behavior dramatically separated during cue light illumination. Paired rats increased freezing and locomotion during cue illumination, then increased locomotion, backpedaling, and jumping during and following shock presentation. At the same time, paired rats decreased rearing and food cup behavior. By contrast, unpaired rats increased light scaling, rearing, and light rearing during cue light illumination. Although increases in light rearing were observed in paired rats, these increases were blunted compared to unpaired rats. Trial-by-trial ethograms are shown in *Figure 3—figure supplement 1*.

In support of group differences across all behaviors, MANOVA for all 12 behaviors [factors: time (87, 200-ms bins), group (paired vs. unpaired), intensity (0.50 mA vs. 0.35 mA), and sex (female vs. male)] revealed a significant group × time interaction ($F_{1032,24768} = 2.50$, $p=7.41 \times 10^{-124}$). MANOVA additionally revealed a significant group × time × sex interaction ($F_{1032,24768} = 1.13$, $p=0.003$) and a significant group × time × intensity interaction ($F_{1032,24768} = 1.64$, $p=5.47 \times 10^{-33}$), which we return to in our univariate analyses.

To reveal behavior-specific differences between groups, we performed univariate ANOVA using a Bonferroni-corrected p-value of 0.004167 (0.05/12) to reduce type 1 error. Univariate ANOVA (factors same as MANOVA) found significant group × time interactions for 8 of the 12 behaviors: freeze ($F_{86,2064} = 2.13$, $p=1.85 \times 10^{-8}$; *Figure 3C*), locomote ($F_{86,2064} = 3.72$, $p=5.04 \times 10^{-26}$; *Figure 3D*), backpedal ($F_{86,2064} = 14.95$, $p=2.34 \times 10^{-159}$; *Figure 3E*), jump ($F_{86,2064} = 3.23$, $p=2.98 \times 10^{-20}$; *Figure 3F*), light scale ($F_{86,2064} = 1.53$, $p=0.003$; *Figure 3G*), rear ($F_{86,2064} = 1.75$, $p=3.79 \times 10^{-5}$; *Figure 3H*), light rear ($F_{86,2064} = 1.47$, $p=0.004$; *Figure 3I*), and cup ($F_{86,2064} = 2.05$, $p=8.75 \times 10^{-8}$; *Figure 3J*). The eight behaviors differentially expressed by paired and unpaired rats varied in direction and timing. Paired rats showed greater freezing during the cue light period, greater locomotion during the cue and post-cue light periods, and greater backpedaling and jumping during the post-cue light period. Unpaired rats showed greater light scaling only during the cue light period, but greater rearing, light rearing, and food cup behavior during the cue and post-cue light periods.

Univariate ANOVA (factors same as above) further found significant group × time × intensity interactions for 3 of the 12 behaviors: locomote ($F_{86,2064} = 1.52$, $p=0.002$), jump ($F_{86,2064} = 2.58$, $p=5.34 \times 10^{-13}$), and backpedal ($F_{86,2064} = 1.63$, $p=2.64 \times 10^{-63}$). Ethograms separating 0.50 mA and 0.35 mA rats are shown in *Figure 3—figure supplement 2*. Paired rats receiving the 0.50 mA foot shock predominantly jumped and locomoted following shock presentation. Paired rats receiving the 0.35 mA foot shock predominantly backpedaled then locomoted following shock presentation. Unpaired rats did not jump, locomote, or backpedal at either shock intensity.

Finally, univariate ANOVA found significant group × time × sex interactions for 3 of the 12 behaviors: rear ($F_{86,2064} = 1.80$, $p=1.41 \times 10^{-5}$), jump ($F_{86,2064} = 1.93$, $p=1.12 \times 10^{-6}$), and backpedal ($F_{86,2064} = 1.63$, $p=2.79 \times 10^{-4}$). Line graphs separately plotting female and male rats for each of the three behaviors are shown in *Figure 3—figure supplement 3*. Sex differences were not all or none – both paired females and males reduced rearing and increased jumping and backpedaling. Instead, differences amounted to timing and magnitude. Paired female rats suppressed rearing during late cue illumination, while males suppressed post-cue. Paired female rats had a longer jump bout, on average, compared to males following shock presentation. Paired males showed greater levels of backpedaling compared to females following shock presentation.

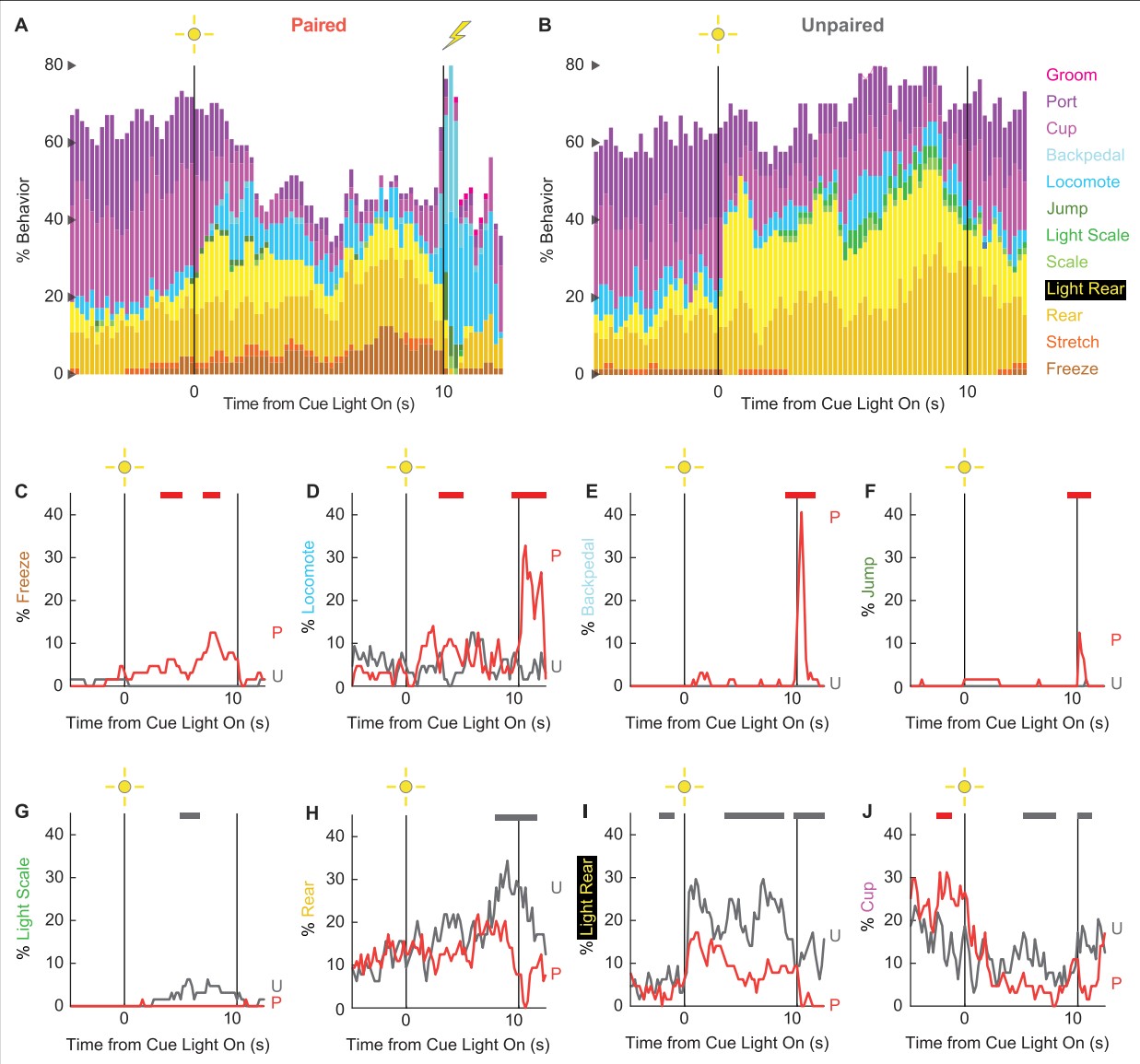

**Figure 3.** Conditioning ethograms. (**A**) Ethogram for paired rats shows % behavior (y axis) for each behavioral category in 200-ms intervals from 5 s prior to cue onset (time 0) to 2.5 s following cue offset (time 10). Paired rats had shock delivered at cue offset. Colors for each behavior category: freeze (light brown), stretch (orange), rear (mustard), light rear (yellow), scale (light green), light scale (green), jump (dark green), locomote (cyan), backpedal (sky blue), cup (purple), port (dark purple), and groom (magenta). (**B**) Ethogram for unpaired rats, details identical to paired rats. Unpaired rats did not receive shock at cue offset. Line graphs for the eight behaviors showing a significant group × time interaction are shown in panels (**C–J**). % behavior is shown for paired rats (P, red) and unpaired rats (U, gray). X axis same as in (**A/B**), y axis is scaled to best visualize behavior patterns. Colored bars at the top of each axis indicate 1 s time periods in which paired and unpaired % behavior differed (independent samples $t$-test, $p<0.05$). Red bars indicate greater behavior in paired rats, while gray bars indicate greater behavior in unpaired rats.

The online version of this article includes the following figure supplement(s) for figure 3:

**Figure supplement 1.** Conditioning ethograms by trial and intensity.

**Figure supplement 2.** Conditioning ethograms by intensity.

**Figure supplement 3.** Conditioning line graphs by sex.

## Behavioral diversity and locomotion persist in extinction

To determine if any behaviors were differentially expressed in the absence of foot shock, we constructed complete ethograms for paired (*Figure 4A*) and unpaired (*Figure 4B*) rats during extinction testing. As during conditioning baseline behavior largely consisted of food-related behavior and rearing, and was similar for paired and unpaired rats. Paired and unpaired behavior diverged following cue light

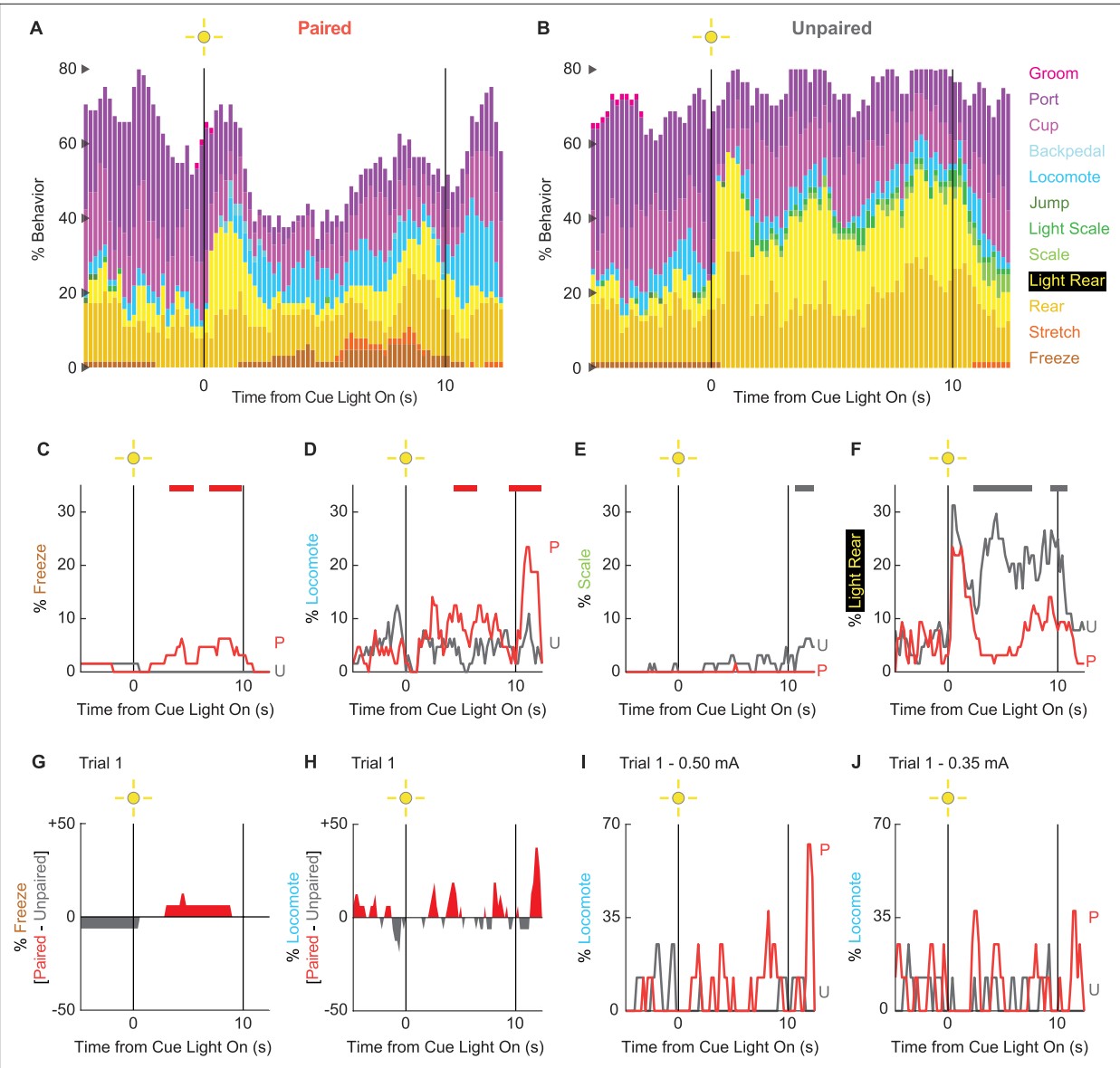

**Figure 4.** Extinction ethograms. (**A**) Ethogram for paired rats shows % behavior (y axis) for each behavioral category in 200-ms intervals from 5 s prior to cue onset (time 0) to 2.5 s following cue offset (time 10). No shock was delivered during this session. Colors for each behavior category: freeze (light brown), stretch (orange), rear (mustard), light rear (yellow), scale (light green), light scale (green), jump (dark green), locomote (cyan), backpedal (sky blue), cup (purple), port (dark purple), and groom (magenta). (**B**) Ethogram for unpaired rats, details identical to paired rats. Line graphs for the four behaviors showing a significant group × time interaction are shown in panels (**C–F**). % behavior is shown for paired rats (P, red) and unpaired rats (U, gray). Colored bars at the top of each axis indicate 1-s time periods in which paired and unpaired % behavior differed (independent samples t-test, p<0.05). Red bars indicate greater % behavior in paired rats, while gray bars indicate greater % behavior in unpaired rats. (**G**) Mean % differential freezing (paired – unpaired) is shown in 200-ms intervals from 5 s prior to cue onset to 2.5 s following cue offset. Areas above zero indicate times when paired behavior exceeded unpaired behavior (red). Areas below zero indicate times when unpaired behavior exceeded paired behavior (gray). (**H**) Mean % differential locomote, shown as in (**G**). Trial 1 locomote line graphs are shown for (**I**) rats receiving 0.50 mA foot shock and (**J**) rats receiving 0.35 mA foot shock. Note that the y axis maximum for (**I**) and (**J**) is twice as large as for (**D**).

The online version of this article includes the following figure supplement(s) for figure 4:

**Figure supplement 1.** Extinction ethograms by trial and intensity.

**Figure supplement 2.** Extinction ethograms by intensity.

**Figure supplement 3.** Trial-by-trial differential behavior.

illumination. Paired rats increased freezing and locomotion during cue illumination, then increased locomotion following cue offset. Unpaired rats increased scaling throughout cue illumination and maintained high levels of light rearing throughout cue and post-cue light periods. Both scaling and light rearing were suppressed in paired rats. Trial-by-trial ethograms are shown in *Figure 4—figure supplement 1*.

In support of group differences across all behaviors, MANOVA for all 12 behaviors [factors: time (87, 200-ms bins), group (paired vs. unpaired), intensity (0.50 mA vs. 0.35 mA), and sex (female vs. male)] revealed a significant group × time interaction ($F_{1032,24768}$ = 1.50, p=4.80 × 10$^{-22}$). MANOVA additionally revealed a significant group × time × intensity interaction ($F_{1032,24768}$ = 1.21, p=1.21 × 10$^{-6}$) but no significant group × time × sex interaction ($F_{1032,24768}$ = 0.98, p=0.70). Revealing behavior-specific differences, univariate ANOVA (factors same as MANOVA) found significant group x time interactions for 4 of the 12 behaviors: freeze ($F_{86,2064}$ = 2.13, p=1.85 × 10$^{-8}$; *Figure 4C*), locomote ($F_{86,2064}$ = 3.72, p=5.04 × 10$^{-26}$; *Figure 4D*), scale ($F_{86,2064}$ = 1.53, p=0.003; *Figure 4E*), and light rear ($F_{86,2064}$ = 1.47, p=0.004; *Figure 4F*). Paired rats showed greater freezing only during the cue light period, as well as greater locomotion during the cue and post-cue light periods. Unpaired rats showed greater scaling and light rearing during the cue and post-cue light periods.

Univariate ANOVA found a significant group × time × intensity interaction only for rearing ($F_{86,2064}$ = 2.32, p=3.87 × 10$^{-4}$). Ethograms separating 0.50 and 0.35 mA rats are shown in *Figure 4—figure supplement 2*. Paired rats receiving the 0.50 mA foot shock suppressed rearing during cue illumination, while 0.35 mA rats did not.

## Locomotion is observed during the first extinction trial

Our rationale for limited extinction trials was to reveal cue-elicited behaviors acquired in the prior conditioning sessions. Given our rationale, the behavior of greatest interest occurs on trial 1. Visually evoked behaviors occurring on trial 1 are not influenced by shock omission (following cue presentation). Our trial 1 analyses focus on freezing and locomotion, the two most prominent cue-elicited behaviors. ANOVA for trial 1 freezing [time (87, 200-ms bins), group (paired vs. unpaired), intensity (0.50 mA vs. 0.35 mA), and sex (female vs. male)] returned a significant group × time interaction ($F_{86,2064}$ = 1.38, p=0.013) but no further group interactions (all Fs < 1.3, all ps>0.09). Paired rats increased freezing over unpaired rats during cue light illumination (*Figure 4G*). ANOVA for trial 1 locomotion (same factors as for freezing) returned a significant group × time interaction ($F_{86,2064}$ = 1.32, p=0.027), but now also a significant group × time × intensity interaction ($F_{86,2064}$ = 1.55, p=0.001). Paired rats at both shock intensities increased locomotion during the cue and post-cue periods (*Figure 4H*). Still, these increases were more prominent for rats receiving the 0.50 mA foot shock (*Figure 4I*) than the 0.35 mA foot shock (*Figure 4J*). Qualitatively, differential freezing (greater freezing in paired rats) accounted for lesser % behavior but stretched over a longer cue light illumination period. Differential locomotion (greater locomotion in paired rats) accounted for greater % behavior, was more transient, and was observed during cue light illumination, as well as following light offset.

## Ethograms predict fear conditioning status

If there is a robust, bidirectional relationship between fear conditioning status and behavior, it should be possible to use ethogram data to predict fear conditioning status (paired vs. unpaired). To test this, we turned to linear discriminant analysis. We used ethogram data to create a linear array for each of the 64 sessions (32 conditioning and 32 extinction). We trained then tested linear classifiers to predict paired vs. unpaired group membership based on the ethogram data. The data of primary interest was mean ± SEM group classification accuracy, with chance classification being 50%.

We first performed linear discriminant analysis for the total ethogram data (1044 value array for each session, 12 behaviors × 87 samples). We compared linear discriminant analysis for the total ethogram data to two kinds of shuffled ethogram data. Session-shuffled ethogram data were shuffled by row, meaning that the relationship between group membership and ethogram was lost. Temporal-shuffled ethogram data were shuffled by column, meaning the group membership information was intact, but the temporal structure of behavior was lost.

Ethograms predicted group membership, and prediction did not deteriorate when temporal structure was lost. Mean ± SEM classification accuracy for the total ethogram data was 82.73 ± 0.003%, exceeding chance (one-sample *t*-test, p=3.87 × 10$^{-102}$; *Figure 5A*). Mean ± SEM classification accuracy

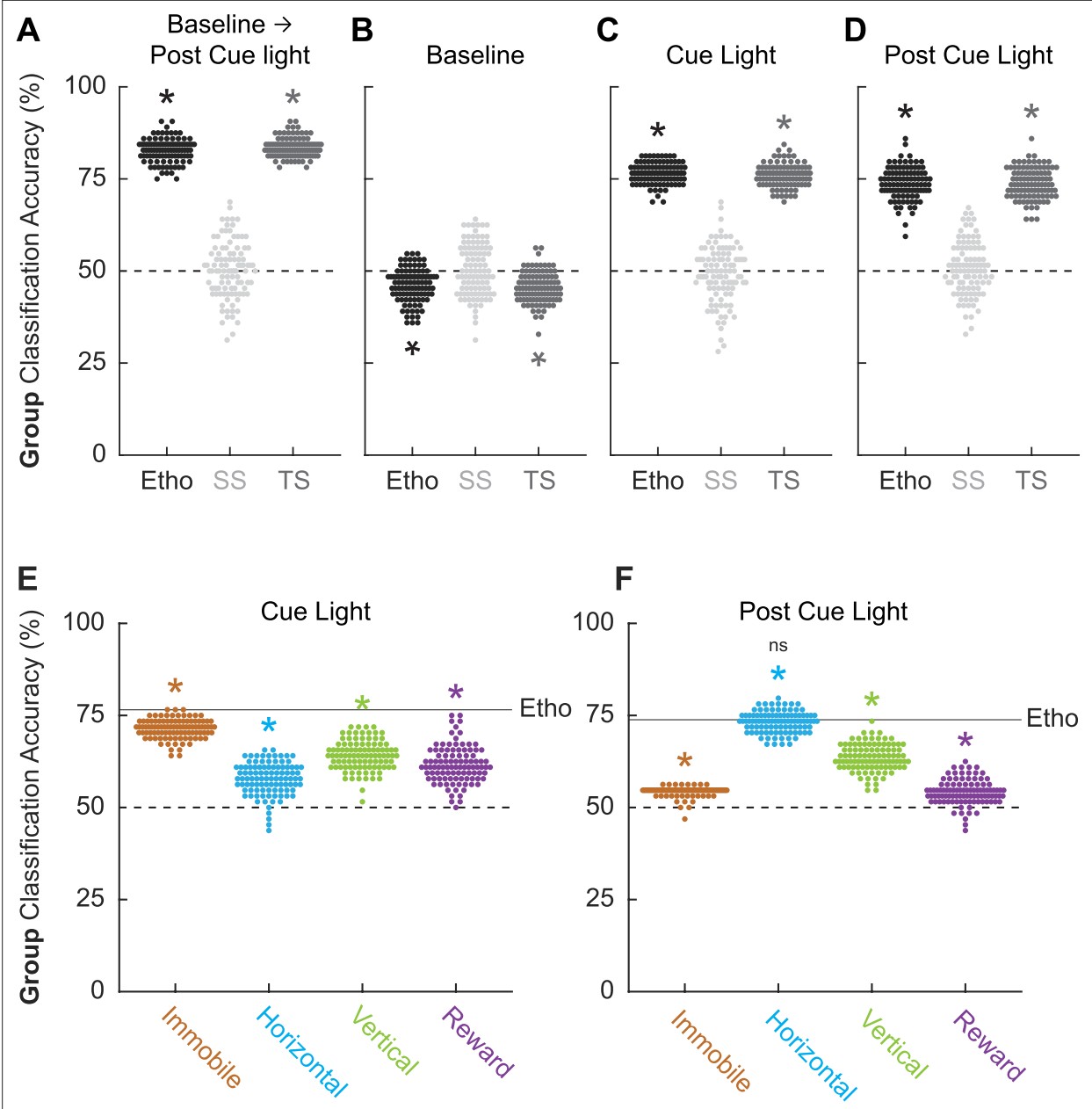

**Figure 5.** Linear discriminant analysis results. Group classification accuracy (paired vs. unpaired) is shown for (**A**) all frames (baseline → post-cue light) comprising the ethogram data, (**B**) baseline, (**C**) cue light illumination, and (**D**) post-cue light illumination. Ethogram data were intact (Etho, black), shuffled by session (SS, light gray), or shuffled temporally (TS, dark gray). Each data point represents the accuracy of a single model. The dotted line indicates chance classification. Group classification accuracy for (**E**) cue light illumination is shown for separate, multibehavior categories (Immobile, light brown; Horizontal, cyan; Vertical, green; and Reward, purple). The dotted line indicates chance classification, while the solid line indicates mean classification accuracy for the total ethogram data during cue light illumination. Group classification accuracy for (**F**) post-cue light illumination is shown for separate, multibehavior categories (color as in **B**). The dotted line indicates chance classification, while the solid line indicates mean classification accuracy for the total ethogram data post-cue light illumination. *Significance of a one-sample t-test compared to 50% (chance). nsNonsignificance of an independent samples t-test comparing horizontal classification to total ethogram classification.

for session-shuffled data was 50.23 ± 0.008%, which did not differ from chance (one-sample t-test, p=0.76). Mean ± SEM classification accuracy for the temporal-shuffled ethogram data was 83.53 ± 0.003%, exceeding chance (one-sample t-test, p=2.77 × 10⁻¹¹¹). Total ethogram and temporal-shuffled ethogram classification each exceeded session-shuffled classification (independent samples t-test, both ps<1 × 10⁻¹⁰) but did not differ from one another (independent samples t-test, p=0.074).

To determine which trial components provided the most group information, we broke the 87 frames into baseline (frames 1–25, *Figure 5B*), cue light illumination (frames 26–75, *Figure 5C*), and post-cue light (frames 76–87, *Figure 5D*) periods. Group membership could not be classified from baseline ethogram data. Indeed, group classification was below chance for the total ethogram data (46.11 ± 0.005%; one-sample *t*-test, p=$1.19 \times 10^{-13}$) and the temporal-shuffled ethogram data (45.53 ± 0.004%; one-sample *t*-test, p=$2.30 \times 10^{-18}$). Group membership was readily classified during the cue light illumination period [total ethogram data (76.53 ± 0.003%; one-sample *t*-test, p=$1.48 \times 10^{-98}$); temporal-shuffled ethogram data (75.98 ± 0.003%; one-sample *t*-test, p=$4.24 \times 10^{-94}$)] as well as during the post-cue light period [total ethogram data (73.83 ± 0.004%; one-sample *t*-test, p=$1.52 \times 10^{-75}$); temporal-shuffled ethogram data (73.81 ± 0.004%; one-sample *t*-test, p=$1.30 \times 10^{-77}$)]. Total cue ethogram and temporal-shuffled cue ethogram classification each exceeded session-shuffled classification (independent samples *t*-test, both ps<$1 \times 10^{-10}$) but did not differ from one another (independent samples *t*-test, both ps>0.2). The same *t*-test significance pattern was obtained for post-cue group classification. Observing comparable classification during the cue and post-cue light periods means that group classification cannot be simply attributed to shock responding.

## Total ethogram classification is superior to behavior subsets

Finally, we asked if comparable group classification could be achieved using behavior category subsets. To test this, we divided the ethogram into four multibehavior categories: Immobile (freeze and stretch), Horizontal (locomote and backpedal), Vertical (rear, scale, jump, light rear, and light scale), and Reward (cup and port). Group classification during cue light illumination (*Figure 5E*) was greater than chance for each multibehavior category, but was variable across categories: Immobile (mean ± SEM group classification, 71.39 ± 0.003%; one-sample *t*-test compared to 50%, p=$7.35 \times 10^{-93}$), Horizontal (57.80 ± 0.004%; p=$1.34 \times 10^{-32}$), Vertical (64.17 ± 0.004%; p=$3.56 \times 10^{-58}$), and Reward (61.59 ± 0.005%; p=$5.75 \times 10^{-42}$). Critically, group classification based on the complete ethogram was superior to each category subset: (independent samples *t*-test, Ethogram vs. Immobile, p=$5.93 \times 10^{-25}$; Ethogram vs. Horizontal, p=$3.66 \times 10^{-58}$; Ethogram vs. Vertical, p=$4.10 \times 10^{-42}$; Ethogram vs. Reward, p=$1.08 \times 10^{-45}$).

Group classification post-cue light illumination (*Figure 5F*) was also greater than chance for each multibehavior category, but variable across categories. Relative to the cue light period, the Immobile category performed worse during the post-cue light period (mean ± SEM group classification, 54.36 ± 0.0013%; one-sample *t*-test compared to 50%, p=$2.84 \times 10^{-55}$), while the Horizontal category performed better (71.39 ± 0.0028%; p=$4.27 \times 10^{-93}$). The Vertical category performed similarly during the post-cue light period (63.58 ± 0.0036%; p=$2.99 \times 10^{-60}$), while the Reward category performed worse (54.19 ± 0.0034%; p=$4.63 \times 10^{-22}$). Similar to the cue period, group classification for the post-cue period based on the complete ethogram was superior to each category subset except for Horizontal behaviors: (independent samples *t*-test, Ethogram vs. Immobile, p=$5.07 \times 10^{-66}$; Ethogram vs. Horizontal, p=0.29; Ethogram vs. Vertical, p=$3.85 \times 10^{-31}$; Ethogram vs. Reward, p=$2.42 \times 10^{-76}$). Group classification during and following cue light illumination was best achieved using the total ethogram data, rather than a behavior category subset.

## Discussion

Here we show that a shock-paired visual cue elicits diverse behaviors. Among these behaviors, we observed low levels of freezing. Extending our prior results, we observed robust locomotion, which spiked around the time of expected foot shock. Fear-conditioned locomotion was sensitive to shock intensity and was observed in both sexes. Finally, we show that ethograms can predict whether a rat had paired or unpaired cue and shock presentations. Ethograms spanning all 12 quantified behaviors surpassed predictions based on a subset of behaviors, including immobile behaviors: freezing and stretching.

We have previously observed diverse behaviors elicited by a fear-conditioned auditory cue, including locomotion (*Chu et al., 2024*). In addition to utilizing conditioned suppression (*Estes and Skinner, 1941*), those experiments used less common designs (*Berg et al., 2014*). Rats underwent multi-cue discrimination consisting of danger and safety cues (CS+ vs. CS-) or danger, uncertainty, and safety cues (deterministic and probabilistic cue-shock relationships). Rats received 12–16 fear

conditioning sessions and up to 96 shocks. This number of shocks is comparable to overtraining procedures (*Maren, 1998*), in which compensatory brain and behavioral responses can be observed following brain manipulations such as neurotoxic lesions of the basolateral amygdala (*Zimmerman et al., 2007*). Thus, it is possible that extended training produces fear behaviors that are not observed with limited training. Finally, rats were presented with more complex auditory cues than commonly employed pure tones (*LeDoux et al., 1990*) or white noise (*Maren et al., 2001*). Design complexity, extended training, and stimulus complexity may have acted independently or combined to produce behavioral diversity and fear-conditioned locomotion. However, these factors do not account for our present findings. Behavioral diversity and robust fear-conditioned locomotion can be observed in rats receiving limited visual cue/shock pairings.

Our current and prior studies corroborate findings from other laboratories (*Gruene et al., 2015*; *Gruene et al., 2015*; *Fadok et al., 2017*) observing locomotion to a shock-paired cue, but are notable for their simplicity and associative basis. Using a serial compound conditioning procedure (tone → white noise → shock), Le and colleagues found robust flight to white noise presentation (*Le et al., 2024*). Robust flight was only observed when the serial compound was paired with shock. Totty and colleagues also observed that white noise elicited flight whether it was the first or second compound stimulus (*Totty et al., 2021*). Conditioned flight to white noise in a serial compound may be a stimulus-specific instance of flight. Gruene and colleagues first demonstrated darting to a shock-paired auditory cue (*Gruene et al., 2015*). Darting has since been demonstrated in additional studies (*Mitchell et al., 2022*; *Mitchell et al., 2024*). Although darting in their rats is comparable to loco-motion in our paired rats, the associative nature of darting has yet to be demonstrated via the use of an unpaired control. Observing robust fear-conditioned locomotion in a simple, associative setting provides a strong behavioral basis to uncover neural circuits for this essential but overlooked fear behavior (*Borkar et al., 2024*).

Behavioral neuroscience (*Hsu and Yttri, 2021*; *Iglesias et al., 2023*; *Luxem et al., 2023*; *Goodwin et al., 2024*) and Pavlovian fear conditioning (*Cai et al., 2020*; *Chanthongdee et al., 2024*) are moving toward machine scoring of behavior. Unsupervised machine scoring may reveal behaviors not previously observed in response to a fear-conditioned cue (*Chanthongdee et al., 2024*) as it does not rely on assumptions made about which behaviors should be present or absent (*Hsu and Yttri, 2021*). Supervised machine scoring, such as convolutional neural networks (*Wu et al., 2020*), trains a model based on hand-scored human data. A supervised scoring approach is useful when known behaviors – such as freezing and locomotion – are firmly established in a behavioral setting. Scored frames from the current experiments, as well as our prior study (*Chu et al., 2024*), are free to download via repository explicitly for these purposes. Machine scoring holds promise to make comprehensive ethograms – like in our studies – more prevalent in the field.

Comprehensive ethograms are necessary to reveal the neural basis of *Pavlovian fear conditioning*, as opposed to the neural basis of conditioned freezing. This is because Pavlovian fear conditioning is a collection of behaviors and processes rather than a single behavior or process (*McDannald, 2023*). If freezing was the totality of Pavlovian fear conditioning, as expressed through overt behavior, we should observe two results. First, freezing should be specific to a shock-paired cue. We observe this. Second, paired versus unpaired fear conditioning status should be predicted equally well by freezing alone vs. freezing + all other behaviors. We do not observe this for freezing or *any* behavior. This means that focusing on any individual behavior will miss information about fear conditioning status. A stronger, bidirectional relationship between fear conditioning status and behavior is observed when all behaviors present are considered.

Here we show that a shock-paired visual cue elicits diverse behaviors, including locomotion and freezing. Our results solidify and extend a directional relationship from Pavlovian fear conditioning → behavior. We further demonstrate that the strongest, reverse directional relationship (behavior → Pavlovian fear conditioning) is obtained when all behaviors present are considered. Observing diverse behaviors within a single conditioning procedure has stronger parallels to the diverse symptoms that define anxiety disorders. Comprehensive behavioral approaches, as performed here, will be essential in revealing more complete neural bases for Pavlovian fear conditioning, with greater relevance to disordered anxiety.

## Materials and methods

### Subjects

Experiments 1 and 2 each used 32 adult Long Evans rats (16 females), for a total of 64 rats (32 females). Rats were obtained from Charles River Laboratories on postnatal day 55. Rats were single-housed on a 12 hr light cycle (lights off at 6:00 pm) and maintained at their initial body weight with standard laboratory chow (18% Protein Rodent Diet #2018, Harlan Teklad Global Diets, Madison, WI). Water was available ad libitum in the home cage. All experiments were carried out in accordance with the NIH guidelines regarding the care and use of rats for experimental procedures. All procedures were approved by the Boston College Animal Care and Use Committee. The Boston College experimental protocol supporting these procedures is 2024-001.

### Behavior apparatus

The behavior apparatus consisted of eight individual chambers with aluminum front and back walls, clear acrylic sides and top, and a grid floor. LED strips emitting 940 nm light were affixed to the acrylic top to constantly illuminate the behavioral chamber for frame capture. 940 nm illumination was chosen because rats do not detect light wavelengths exceeding 930 nm (*Nikbakht and Diamond, 2021*). An array of three infrared lights were mounted to the chamber door to track the trial period (baseline, light, and post-light). Each grid floor bar was electrically connected to an aversive shock generator (Med Associates, St. Albans, VT). An external food cup, a central port equipped with infrared photocells, and a key light were present on one wall.

### Pellet exposure and nose poke shaping

Identical pellet exposure and nose poke shaping were performed for Experiments 1 and 2. Rats were food-restricted and specifically fed to maintain their body weight throughout behavioral testing. Each rat was given four grams of experimental pellets in their home cage in order to overcome neophobia. Next, the central port was removed from the experimental chamber, and rats received a 30-min session in which one pellet was delivered every minute. The central port was returned to the experimental chamber for the remainder of behavioral testing. Each rat was then shaped to nose poke in the central port for experimental pellet delivery using a fixed ratio schedule in which one nose poke into the port yielded one pellet. Shaping sessions lasted 30 min or until approximately 50 nose pokes were completed. Each rat then received five sessions during which nose pokes into the port were reinforced on a variable interval schedule. Session 1 used a variable interval 30 s schedule (poking into the port was reinforced every 30 s on average). All remaining sessions used a variable interval 60 s schedule. For the remainder of behavioral testing, nose pokes were reinforced on a variable interval 60 s schedule independent of light illumination and shock presentation.

### Light pre-exposure

Identical light pre-exposure was performed for Experiments 1 and 2. Each rat was pre-exposed to the 10-s light, four times during two, 43 min sessions (Monday and Tuesday). The two pre-exposure sessions meant that each rat received eight total, light pre-exposures. Although light intensity was greatest at the source, light illumination generally illuminated the experimental chamber, which was completely dark to the rat at all other times.

### Light illumination and shock presentation

Following pre-exposure, the 32 rats in each experiment were evenly divided into two groups: paired (16 rats, 8 females) and unpaired (16 rats, 8 females). Each paired and unpaired rat received four light and four shock presentations during two, 43 min sessions (Wednesday and Thursday). The two sessions meant that each rat received eight total light and shock presentations. For rats in the paired group, the light was presented for 10 s. Light offset precisely coincided with foot shock presentation (0.5 s). For rats in the unpaired group, light (10 s) and shock (0.5 s) were presented at distant times. The inter-event interval (in which both light and shock are considered events) was greater than 5 min. The order of unpaired light and shock presentation was randomly determined and differed for each rat, each session.

## Foot shock intensity

For both experiments, two levels of foot shock intensity were used. For Experiment 1, half of the rats received a 0.25 mA foot shock and half a 0.15 mA foot shock. The total experimental design yielded four groups: paired 0.25 (eight rats, four females), paired 0.15 (eight rats, four females), unpaired 0.25 (eight rats, four females), and unpaired 0.15 (eight rats, four females). For Experiment 2, half of the rats received a 0.50 mA foot shock and half a 0.35 mA foot shock. The total experimental design yielded four groups: paired 0.50 (eight rats, four females), paired 0.35 (eight rats, four females), unpaired 0.50 (eight rats, four females), and unpaired 0.35 (eight rats, four females).

## Extinction test

Identical extinction tests were performed for Experiments 1 and 2. The single extinction session (Friday) was similar to the pre-exposure session. Each rat received four, 10-s light illuminations in a 43 min session.

## Calculating suppression ratio

Time stamps for cue light illumination, shock presentation, and nose pokes (photobeam break) were automatically recorded using the Med Associates program. The baseline nose poke rate was calculated for each trial by counting the number of pokes during the 20-s pre-light period and multiplying by 3. The light nose poke rate was calculated for each trial by counting the number of pokes during the 10-s light period and multiplying by 6. Nose poke suppression was calculated as a ratio: (baseline poke rate – light poke rate)/(baseline poke rate + light poke rate). A suppression ratio of '1' indicated complete suppression of nose poking during cue light illumination relative to baseline. A suppression ratio of '0' indicates equivalent nose poke rates during baseline and light presentation. Gradations in suppression ratio between 1 and 0 indicated intermediate levels of nose poke suppression during cue light illumination relative to baseline. Negative suppression ratios indicated increased nose poke rates during cue light illumination relative to baseline.

## Frame capture system

Behavior frames were captured using Imaging Source monochrome cameras (DMK 37BUX28; USB 3.1, 1/2.9″ Sony Pregius IMX287, global shutter, resolution 720 × 540, trigger in, digital out, C/CS-mount). Frame capture was triggered using the Med Associates behavior program. The 28 V Med Associates pulse was converted to a 5 V TTL pulse via Adapter (SG-231, Med Associates). The TTL adapter was wired to the camera's trigger input. Captured frames were saved to a PC (OptiPlex 7470 All-in-One) running IC Capture software (Imaging Source). Frame capture began precisely 5 s before light onset, continued throughout 10-s cue light illumination, and ended 2.5 s following light offset. For paired rats, this meant that the frames were captured during the 0.5 s shock and the 2 s following. For unpaired rats, the empty 2.5 s period was captured. Frames were captured at a rate of 5 per second, with a target of capturing 87 frames per trial, and 348 frames per session.

## Anonymizing trial information

A total of 256 trials were scored from the 32 rats in Experiment 2. Trials came from the second session of light/shock presentations (Thursday) and the extinction test (Friday). We anonymized trial information in order to score behavior without bias. The numerical information from each trial (rat #, session #, and trial #) was encrypted as a unique number sequence. A unique word was prepended to the sequence. The result was that each trial was converted into a unique word + number sequence. For example, trial dw01_01_03 (rat #1, session #1, and trial #3) would be encrypted as abundant28515581. The trials were randomly assigned to six observers. The result of trial anonymization was that observers were completely blind to the subject, group, and session. Further, random assignment meant that different observers scored the four trials composing a single session.

## Post-acquisition frame processing

A MATLAB script sorted the 348 frames from the four session trials into four folders, one for each trial, each containing 87 frames. Each 87-frame trial was made into an 87-slide PowerPoint presentation to be used for hand scoring.

## Box 1. Behavior definitions

**Background**
Specific behavior cannot be discerned because the rat is turned away from the camera or position of forepaws is not clear, or because the rat is not engaged in any of the other behaviors.

**Backpedal**
Rapid backward displacement of the body. Often (not always), the head is down and/or the back is arched. The backward component of movement must be stronger than the lateral component. A lateral movement is not a backpedal. Backpedal considers the current frame ($t$) and the next two frames ($t + 1$ and $t + 2$). By frame $t + 2$ the body and both back feet must be displaced backward relative to frame $t$. The rat can move the body and both feet by $t + 1$, $t + 2$, or move in a combination of both frames; all count as backpedal for trial $t$. (See illustration for reference.)

**Cup**
Any part of the nose above the food cup but below the nose port.

**Freeze**
Arched back and stiff, rigid posture in the absence of movement, all four limbs on the floor (often accompanied by hyperventilation and piloerection). Side-to-side head movements and up-and-down head movements that do not disturb rigid posture are permitted. Activity such as sniffing, or investigation of the bars is not freezing. Freezing, as opposed to pausing, is likely to be three or more frames (600+ ms) long.

**Groom**
Any scratching, licking, or washing of the body.

**Jump**
All four limbs are off the floor. Includes hanging, which is distinguished when hind legs are hanging freely.

**Locomote**
Propelling body across the chamber on all four feet, as defined by the movement of back feet. Movement of back feet with front feet off the floor is rearing. Locomote considers the current frame ($t$) and the next two frames ($t + 1$ and $t + 2$). By frame $t + 2$ the body and both back feet must be displaced forward relative to frame $t$. The rat can move the body and both feet by $t + 1$, $t + 2$, or move in a combination of both frames; all count as locomote for trial $t$.

**Port**
Any part of the nose in the port. Often standing still in front of the port but sometimes tilting head sideways with the body off to the side of the port.

**Rear**
One or two hind legs on the grid floor with both forepaws off the grid floor and not on the food cup. Usually (not always) stretching to the full extent, forepaws usually (not always) on top of side walls of the chamber, often pawing walls; may be accompanied by sniffing or slow side-to-side head movement. It does not include grooming movements or eating, even if performed while standing on hind legs.

**Light rear**
A rear happens within the 3D space defined by these planes: the nose is above the poke (X), between the metal columns in the light area (Y), within the depth of the front edge of the food cup (Z). If the rat's body is outside the columns BUT the nose is within the defined 3D space, it is a Light Rear. Often, the body will be near the wall, and/or the forepaws will be on the wall. The nose can be above the light. The light does not have to be on to be scored as a Light Rear.

**Scale**
All four limbs are off the floor but at least two limbs are on the side of the chamber. Standing on the food cup counts as scaling.

> **Light scale**
> A scale happens when the nose is above the poke and between the metal columns in the light area. Including above the light. The light does not have to be on to be scored as a light scale.
> **Stretch**
> The body is elongated with the back posture 'flatter' than normal. Stretching is often accompanied by immobility, like freezing, but is distinguished by the shape of the back.

## Behavior categories and definitions

Frames were scored as one of 12 behavior categories and are defined in *Box 1*.

## Frame scoring system

Frames were scored using a specific procedure. Frames were first watched in real time in Microsoft PowerPoint by setting the slide duration and transition to 0.19 s, then playing as a slideshow. Behaviors clearly observed were noted. Next, the observer went through all the frames, scoring one behavior at a time. A standard scoring sequence was used: groom, port, cup, light rear, rear, light scale, scale, jump, freeze, locomote, backpedal, stretch. When the specific behavior was observed in a frame, that frame was labeled. Once all behaviors had been scored, the video was rewatched for freezing. In our procedure, cup is defined based on the rat's location and freezing is defined based on the rat's behavior. If the two co-occurred, the cup superseded freezing because it was prevalent during the baseline behavior. However, the co-occurrence of cup and freezing was low. The unlabeled frames were then labeled 'background'. Finally, all the background frames were checked to ensure they did not contain a defined behavior.

## Statistical analyses

MANOVA and univariate ANOVA were performed for suppression ratios and specific behaviors. Sex was used as a factor for all analyses. Group, session, intensity, and time were used as factors when relevant. Univariate ANOVA following MANOVA used a Bonferroni-corrected p-value significance of 0.004167 (0.05/12) to account for the 12 quantified behaviors. Post hoc comparisons were made using independent samples and one-sample *t*-tests.

## Linear discriminant analysis

MATLAB's fitcdiscr function was used to predict group membership (paired vs. unpaired) from ethogram data. Linear discriminant analysis (LDA) is a linear classification method that projects the ethogram features into a space that maximizes the separation between groups. Each LDA model used the entire dataset without any initial data splitting. To evaluate the performance of the LDA model, tenfold cross-validation was applied using MATLAB's crossval function. Cross-validation divides the dataset into 10 equal parts (folds). For each iteration, the model was trained on nine folds and tested on the remaining fold. This process was repeated 10 times so that each fold served as a test set once. The accuracy of the model (defined as 1 minus the loss) was assessed by computing the classification error for each fold, and the overall performance was averaged across all folds. The crossval function was used in conjunction with kfoldLoss, which returns the average classification error across the folds. The performance of the LDA model was quantified using the average classification accuracy across the 10 folds. This approach provided a robust estimate of the model's generalization capability to unseen data, reducing the likelihood of overfitting. 100 LDA models were made for each analysis. Prediction accuracies from each model were saved and used to calculate mean ± SEM prediction accuracies across the 100 models for each analysis.

## Additional information

### Competing interests

Michael A McDannald: Reviewing editor, eLife. The other authors declare that no competing interests exist.

## Funding

| Funder | Grant reference number | Author |
|--------|------------------------|--------|
| National Institutes of Health | R01-MH117791 | Michael A McDannald |

The funders had no role in study design, data collection and interpretation, or the decision to submit the work for publication.

## Author contributions

David C Williams, Conceptualization, Data curation, Methodology, Project administration; Amanda Chu, Investigation, Methodology, Writing – review and editing; Nicholas T Gordon, Aleah M DuBois, Suhui Qian, Jacob B Boyce, Liliuokalani H Counsman, Investigation, Methodology; Genevieve Valvo, Selena Shen, Anaise C Fitzpatrick, Investigation; Mahsa Moaddab, Project administration, Writing – review and editing; Emma L Russell, Methodology, Writing – review and editing; Michael A McDannald, Conceptualization, Formal analysis, Funding acquisition, Visualization, Methodology, Writing – original draft, Writing – review and editing

## Author ORCIDs

Michael A McDannald (iD) https://orcid.org/0000-0001-8525-1260

## Ethics

Animal care was in accordance with NIH and Boston College guidelines. The Boston College experimental protocol supporting these procedures is 2024-001.

## Decision letter and Author response

Decision letter https://doi.org/10.7554/eLife.102782.sa1
Author response https://doi.org/10.7554/eLife.102782.sa2

# Additional files

## Supplementary files

MDAR checklist

## Data availability

Raw frames and scored behaviors have been deposited: https://doi.org/10.7910/DVN/Z4YJRJ.

The following dataset was generated:

| Author(s) | Year | Dataset title | Dataset URL | Database and Identifier |
|-----------|------|---------------|-------------|-------------------------|
| McDannald MA | 2025 | Behavior frames and observer judgments from 32 rats (16 females) receiving Pavlovian fear conditioning to a visual cue | https://doi.org/10.7910/DVN/Z4YJRJ | Harvard Dataverse, 10.7910/DVN/Z4YJRJ |

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
