## [Editor Report]

This report provides a useful characterization of the behaviors evoked by a Pavlovian conditioned stimulus (CS) paired with foot shock in male and female rats. The aim of the study was to assess the generalizability of the authors' past findings (Chu et al., 2024), established using an auditory CS (a tone), to a visual CS (a light). The reviewers appreciated the extensive nature of the task undertaken: there was agreement that the methods and analyses used to produce the study findings are solid. This work will be of interest to those who study associative learning, fear, ethological assessment, and behavior more broadly.

---

## [Decision Letter]

**Decision letter after peer review:**

Thank you for submitting your article "Ethograms reveal a fear conditioned visual cue to organize diverse behaviors in rats" for consideration by *eLife*. Your article has been reviewed by 3 peer reviewers, and the evaluation has been overseen by a Reviewing Editor and Kate Wassum as the Senior Editor.

Essential Revisions (for the authors):

The reviewers identified a number of points that, when addressed, will increase the overall significance of the work.

*Reviewer 1:*

1. The work could be improved by providing a description of the behaviors elicited by the weak shocks in experiment 1. That experiment failed to produce conditioned suppression or freezing but did the weak shocks elicit any of the other behaviors observed in experiment 2?

2. I would have also liked to see a description of the behaviors in the first conditioning session of experiment 2 to see how these related to those observed in the well-documented 2nd session.

3. Extinction consisted in a single session, not sufficient to produce a significant loss of freezing, etc. Additional sessions would have been useful to provide an indication of the extent to which the conditioned responses converge with those exhibited by the control groups.

*Reviewer 2:*

1. In Figure 1D, the females in the paired group on the right are clustering higher than the males, and this is also present in Figure 1F, in the left paired condition. I think the authors should increase the N in these conditions to determine if there are significant sex differences.

2. It could be more informative to show early versus late bins for extinction learning.

3. In Figure 3A, it appears that either Port or Cup behavior is increased at the time of shock. Is this correct? If so, why?

4. For the ethograms in Figures3+4, are the data averaged across the session? For extinction data, it might be important to show data from early trials compared to data from late trials of extinction.

5. Figure 1D x-axis appears to be mislabeled, or the legend is incorrect, as the legend indicates that 0.25 mA is signified by diamonds.

6. For me, the color scheme in Figure 2A heatplot is difficult on the eyes. Consider switching to another color scheme such as viridis.

7. The colorful ethogram plots in the figures suggest continuous data collection, but the data were collected in 200-ms intervals. I think a histogram with 200-ms bins might be a better choice for data visualization.

*Reviewer 3:*

1. A more detailed explanation of this study's contributions to the field would help strengthen the paper.

2. An extensive analysis has been done to demonstrate the reliability of the ethogram analysis. This effort is commendable; however, it does not provide information that is readily useful to the scientific community, especially as the field is now shifting toward machine learning approaches. Please comment on this in the revisions.

3. The manuscript presents some typos. For instance, on the second page of the intro "simper" is written instead of "simpler". Additionally, "behavior" is written twice in the next paragraph. Please correct.

Additional

1. Some of the claims in the Introduction (e.g., that a visual CS is known to produce little conditioned freezing) are supported by older references and ignore many recent findings. Please update the references for this section of the Introduction and revise any statements accordingly.

2. Re the method – The authors created four categories of 12 "mutually exclusive" behaviours. The categories included immobility and reward. The implication is that rats cannot have been immobile while in the food cup where the reward was delivered. However, it is not clear at all why this should be true (i.e., rats can and will freeze with their head inside a food cup). Please clarify how these behaviours were recorded in relation to each other.

3. Re inter-rater reliability, was there a particular behaviour or category of behaviours that had a lower/higher inter-rater reliability than the others?

4. Re sex differences: given that the rats were hungry, was body weight included as a covariate that might account for the apparent sex differences?

*Reviewer #1:*

Two experiments are reported but only one is described in detail. The design of each experiment was simple but powerful: one group of rats received pairings of a visual stimulus and foot shock while the other group received unpaired presentation of the stimulus and shock. The other factor was shock intensity (two values in each experiment; very low in experiment 1 and moderate in experiment 2). The aim of the work was to compile a list of the behaviors elicited by the visual stimulus across its paired or unpaired presentations with shock. The compilation of these behaviors required an impressive amount of work and their analyses were equally impressive. The conclusion appeared to be that incorporating all of the behaviors discriminated better than freezing (a common index of threat conditioning) between the paired and unpaired groups. That said, the work yielded little else of theoretical significance.

The work could be improved by providing a description of the behaviors elicited by the weak shocks in experiment 1. That experiment failed to produce conditioned suppression or freezing but did the weak shocks elicit any of the other behaviors observed in experiment 2? I would have also liked to see a description of the behaviors in the first conditioning session of experiment 2 to see how these related to those observed in the well-documented 2nd session. In view of the effort required to provide such a complete assay of threat related behaviors and the fact that they provide marginally more information than freezing, it is unlikely that such an assay will replace freezing in the search for the brain mechanisms underlying threat conditioning, I appreciate that the assay was restricted to the protocols commonly used to produce threat conditioning as assessed by freezing. However, changing the environmental space to allow the rats to move into another chamber when the visual stimulus appeared in the conditioning chamber could be informative with respect to the general idea that defensive behaviors are organized in part by the environmental affordances. Finally, extinction consisted of a single session, not sufficient to produce a significant loss of freezing, etc. Additional sessions would have been useful to provide an indication of the extent to which the conditioned responses converge with those exhibited by the control groups.

*Reviewer #2:*

In the current manuscript, the authors compile detailed ethograms for male and female rats undergoing conditioned suppression of reward seeking. They use 4 different shock levels, and they analyze 12 different behaviors. Only the 2 highest shock levels induced conditioned suppression and these shock levels suppress many appetitive behaviors during the CS. They then use linear discriminant analyses to classify rats as paired or unpaired.

This study is interesting in that it delves into the richness of behavior evident in a simple conditioning task. The results also help explain why conditioned suppression is present in the absence of conditioned freezing behavior. I have a few suggestions that I hope will help strengthen the conclusions and increase the impact of the study.

*Reviewer #3:*

In this study, the authors examine behavioral responses to a visual cue linked to foot shocks within a conditioned suppression paradigm. They demonstrate that conditioned suppression to the visual cue occurs at shock intensities of 0.35 and 0.5, but not at lower levels. An ethogram analysis further shows specific behavioral patterns in response to the visual cue paired with the foot shocks.

Strength of the manuscript:

The authors conducted a comprehensive ethological analysis of behavioral responses to fear-associated cues. Although human annotations may be prone to biases, the authors accounted for this by assessing inter-rater reliability and minimizing rater confusion.

Weaknesses of the manuscript:

This study lacks novelty, as previous research has already shown that a cue paired with footshocks induces freezing and defensive behaviors. It is also unclear what advantage a light cue would offer over a sound cue.

---

## [Author Response]

Essential Revisions (for the authors):The reviewers identified a number of points that, when addressed, will increase the overall significance of the work.

We are grateful for the reviewers time and effort. We have addressed each concern and have extensively revised the manuscript based on their comments.

Reviewer 1:1. The work could be improved by providing a description of the behaviors elicited by the weak shocks in experiment 1. That experiment failed to produce conditioned suppression or freezing but did the weak shocks elicit any of the other behaviors observed in experiment 2?

We initially set out to do exactly this. Holland 1979 showed that freezing linearly scaled with the foot shock intensity and that at low intensities other behaviors emerged because freezing was minimal. Here we did not detect the linear relationship that Holland did. Instead, there was a very clear division between the low foot shock intensities (0.15 mA and 0.25 mA) and high foot shock intensities (0.35 mA and 0.50 mA). This is very likely a product of using a conditioned suppression procedure. The subjects have a reason to be doing something other than expressing defensive behaviors. Namely, working for food. While we think of conditioned suppression as competition in which shock-paired cues influence (and suppress) reward responding, it is likely that the reverse can occur. Indeed, it seems that at low foot shock intensities, the motivation to work for food can eliminate the ability of the shock-paired cue to influence reward seeking.

Your concern also gets at a strength/weakness of conditioned suppression. Conditioned suppression is very good at revealing the summed behavioral properties of shock-paired cues to suppress reward seeking. This is because conditioned suppression is agnostic to which behavior is producing suppression. This makes conditioned suppression an excellent barometer of total behavior when conditioned suppression is observed. But then careful ethograms are needed to reveal exactly how suppression is being achieved. On the flip side, a null conditioned suppression result is very meaningful. If a shock-paired cue is unable to suppress reward behavior, it is very difficult for that cue to also elicit specific defensive behaviors. This is because, by definition of not observing conditioned suppression, any amount of defensive behavior would have to be offset by increased reward behaviors.

In the revised manuscript, we have performed more granular analyses of the conditioned suppression data from Experiments 1 and 2 [Lines 90-127]. These analyses conclusively demonstrate that low shock intensity, paired and unpaired rats do not differ at any point in behavioral testing. Instead, both the 0.25 mA and 0.15 mA rats show a steady decrease in conditioned suppression over the 5 session (20 trials). High shock intensity, paired and unpaired rats show clear and robust divergence during conditioning and in extinction. Because we set out to determine the behaviors elicited by a shock-paired cue that was associative in nature, we chose to construct comprehensive ethograms for only Experiment 2.

2. I would have also liked to see a description of the behaviors in the first conditioning session of experiment 2 to see how these related to those observed in the well-documented 2nd session.

Unfortunately, we do not have behavior frames for those sessions. We agree those data would have been nice to see. However, we do not think that having those data would change the main conclusions of our study. We wanted to identify behaviors that were specific to Paired rats and were the result of conditioning. Doing this means focusing on the trials during which differential responding is observed (via suppression ratio). With the exception of conditioning trial 4 for the rats receiving the 0.50 mA foot shock, the second conditioning session and the extinction session capture all of the relevant trials. In the revised manuscript, we show trial by trial responding for suppression ratio [Figure 1D] to more clearly show why we focused scoring on the last fear conditioning session and extinction session.

3. Extinction consisted in a single session, not sufficient to produce a significant loss of freezing, etc. Additional sessions would have been useful to provide an indication of the extent to which the conditioned responses converge with those exhibited by the control groups.

This is another good point. Extinction testing is traditionally performed for one of two reasons: 1) To reveal what was learned (or what behaviors were acquired) through prior pairings of cues and outcomes or 2) To assess the loss of that responding. For the first approach, very few extinction trials are given. In fact, the very first extinction trial is the most critical for this approach. This is because the first trial is not influenced by shock omission. Whereas every trial after the first is going to reflect both what was learned in the prior sessions and shock omission in extinction. For the second approach, many trials and even sessions are needed. This is necessary to completely reduce cue-elicited responding to pre-learning levels.

In our experiments, we used the first approach. But to do this well, we should have not only shown trial-by-trial responding, but should have specifically shown the behaviors elicited by the cue light on trial 1. The revised manuscript corrects this oversight by analyzing and visualizing behavior for each trial. Critically, we now perform extensive analyses on only the first trial [Lines 231-249]. These analyses do not merely reinforce but extend our findings. Trial by trial analyses show that post-shock locomotion is prominent on trial 1 and is sensitive to shock intensity. So although both 0.50 mA and 0.35 mA paired rats increase locomotion, this increase is significantly greater in 0.50 mA rats [Figure 4I]. Transient locomotion periods accounted for >60% of all behavior. These results – and additional trial by trial supplemental figures – have been added to the revised manuscript.

Reviewer 2:1. In Figure 1D, the females in the paired group on the right are clustering higher than the males, and this is also present in Figure 1F, in the left paired condition. I think the authors should increase the N in these conditions to determine if there are significant sex differences.

This is an astute observation. When mean cue suppression ratio is plotted, females and males in the 0.15 mA Paired conditioned are clustering separately. And in fact, an independent samples t-test confirms that Paired female and male suppression ratios differ for the 0.15 mA condition (t = 13.02, p = 1.26 x 10^-5^). However, three lines of evidence suggest this finding is not meaningful.

The variation exhibited by the 0.15 mA Paired groups is much lower than expected. Standard error means for these two groups were: 0.01875 (female) and 0.00476 (male). For comparison, standard error means for the 0.15 mA Unpaired groups were: 0.1807 (female) and 0.1165 (male). Standard error means for all other groups in this study, and most studies from our laboratory, are consistent with those from the Unpaired groups.Comparing the 0.15 mA female and male Paired groups to each other suggests that meaningful sex differences exist. However, a more complete comparison of the sex x group pattern suggests the opposite.

Female Paired rats (m = 0.37) showed higher mean suppression ratios than female Unpaired rats (m = 0.038). A power analysis found that n=16 rats per group would be required to detect a significant difference. Male Paired rats (m = 0.019) showed *lower* mean suppression ratios than male Unpaired rats (m = 0.361). A power analysis found that n=9 rats per group would be required to detect a significant difference. The problem is twofold. The male Paired vs. Unpaired difference is in the wrong direction. Male Unpaired suppression ratios are comparable to female Paired suppression ratios, and male Paired suppression ratios are comparable to female Unpaired suppression ratios. There is no theoretical or practical basis to think that low foot shock probabilities invert Pavlovian fear conditioning in a sex-specific manner. A more parsimonious explanation is that we happened to sample some high Paired females and low Paired males. Additional samples are unlikely to replicate this pattern. Indeed, a power analysis for Paired vs. Unpaired rats (with sexes collapsed) found that 13,404 rats would be necessary to detect a significant Paired vs. Unpaired sex difference. These analyses strongly suggest that the observed sex difference in the 0.15 mA Paired condition is spurious.

Finally, the low variability observed in the mean suppression ratio data is not supported when trial by trial suppression ratios are plotted [Figure 1C/D]. We have to admit this was a weakness of the initial submission. Given the small number of trials each rat received, it is most appropriate to plot the trial-by-trial data. We have now done that in the revised manuscript. These plots make it clear that at no point during training or test did Paired rats in either the 0.15 or 0.25 mA condition meaningfully diverge from the Unpaired rats [Figure 1C]. In contrast, the Paired vs. Unpaired divergence is clear in both the 0.35 mA and 0.50 mA conditions [Figure 1D].

For these three reasons, we are confident that adding fewer than 13,404 subjects to the 0.15 and 0.25 mA conditions would not alter any of our conclusions.

2. It could be more informative to show early versus late bins for extinction learning.

We agree and have now made figures showing trial by trial ethograms [Figure 4 – Supplemental Figure 1]. We are using extinction to reveal behaviors acquired during the prior conditioning sessions. In this case, the first extinction trial is most critical as it is not influenced by shock omission, which could alter responding on subsequent trials. We now show freezing and locomote data for Trial 1 [Figure 4G-J]. These results show that fear conditioned locomotion is even greater during trial 1, particularly for rats receiving 0.50 mA foot shock.

3. In Figure 3A, it appears that either Port or Cup behavior is increased at the time of shock. Is this correct? If so, why?

In Figure 3A, port (dark purple) is very, very low prior to, during, and following foot shock. Cup (light purple) is also very low around shock although somewhat higher than port. Both port and cup show higher levels during baseline than any period around shock presentation.

We think you might mean Figure 4A? There, port levels towards the end of cue illumination appear closer to baseline levels. In case this is what you meant; we did a deeper statistical dive on port. We performed ANOVA with trial, group, sex, and intensity as factors. We observed a significant trial x time x group interaction (F_258,6192_ = 1.41, *p* = 2.95 x 10^-5^). We show the SPSS graphical outputs (sorry for the color and formatting). But what you can see is that port behavior remained low through cue illumination for trials 1 and 2, but increased towards cue illumination offset on trials 3 and 4. This pattern did not manifest in Unpaired rats. This pattern is consistent with a change in behavior via extinction, showing that reward behavior first returns near cue offset.

While we think this effect is interesting, we do not include it in the manuscript because it is statistically problematic to follow up on all behaviors measured with univariate ANOVA looking for significant interactions.

**Author response image 1. sa2fig1:** 

4. For the ethograms in Figures3+4, are the data averaged across the session? For extinction data, it might be important to show data from early trials compared to data from late trials of extinction.

This is a reasonable request, and we agree this adds significantly to the manuscript. We have now made figures showing complete behavioral ethograms for each trial [Figure 4 – Supplemental Figure 1]. For completeness, we made a trial-by-trial ethogram figure for the second conditioning session [Figure 3 – Supplemental Figure 1].

5. Figure 1D x-axis appears to be mislabeled, or the legend is incorrect, as the legend indicates that 0.25 mA is signified by diamonds.

We apologize for the error and have fixed the legend. We have totally remade this figure to show all 20 cue illumination trials (8 pre-exposure, 8 conditioning and 4 extinction).

6. For me, the color scheme in Figure 2A heatplot is difficult on the eyes. Consider switching to another color scheme such as viridis.

We have switched to the “Hawaii” color map which is part of a suite of colormaps made to be scientifically accurate, color-blind friendly, etc [Figure 2A]. The Hawaii colormap ranges from purples to browns to greens to cyan. In theory, having color throughout should help readers see observer-observer variation. However, there was not a lot of variation between observer pairs.

7. The colorful ethogram plots in the figures suggest continuous data collection, but the data were collected in 200-ms intervals. I think a histogram with 200-ms bins might be a better choice for data visualization.

We agree and have replotted all histograms as bars [Figure 3, Figure 3 – Supplemental Figure 1, Figure 3 – Supplemental Figure 2, Figure 4, Figure 4 – Supplemental Figure 1, and Figure 4 – Supplemental Figure 2].

Reviewer 3:1. A more detailed explanation of this study's contributions to the field would help strengthen the paper.

The discussion has been completely re-written to describe the study’s contribution to the field [Lines 314-382].

2. An extensive analysis has been done to demonstrate the reliability of the ethogram analysis. This effort is commendable; however, it does not provide information that is readily useful to the scientific community, especially as the field is now shifting toward machine learning approaches. Please comment on this in the revisions.

Our discussion now includes a section on machine learning [Lines 352-363]. We disagree that our findings do not provide information that is readily useful to the scientific community. As a part of this study, and our prior study, we have made the total labeled frames from each free to download:

https://doi.org/10.7910/DVN/Z4YJRJ (this submission)

https://doi.org/10.7910/DVN/HKMUUN (Chu et al. 2025 *eLife*)

As of January 2, 2025, the total dataset of labeled frames from the Chu et al. study has been downloaded 1,156 times. Five scientific groups have directly emailed me (McDannald) about using our frames to train supervised machine learning algorithms. We are also using these frames to train convolutional neural networks. We find that CNNs do very well for static behaviors but have a harder time with behaviors that play out across frames (locomotion and freezing). We are continuing to refine our machine learning approach to precisely captures all behaviors of interest.

3. The manuscript presents some typos. For instance, on the second page of the intro "simper" is written instead of "simpler". Additionally, "behavior" is written twice in the next paragraph. Please correct.

We have fixed these typos and carefully read the manuscript to correct additional typos.

Additional1. Some of the claims in the Introduction (e.g., that a visual CS is known to produce little conditioned freezing) are supported by older references and ignore many recent findings. Please update the references for this section of the Introduction and revise any statements accordingly.

This is fair. While auditory cues are more common, visual cues are used in fear conditioning. However, it is still quite rare for visual cues to be used in a conditioned suppression setting. This matters, as freezing is readily obtained to visual and auditory cues in non-conditioned suppression settings. We cite numerous recent findings and clarify the distinction in the introduction [Lines 49-50].

2. Re the method – The authors created four categories of 12 "mutually exclusive" behaviours. The categories included immobility and reward. The implication is that rats cannot have been immobile while in the food cup where the reward was delivered. However, it is not clear at all why this should be true (i.e., rats can and will freeze with their head inside a food cup). Please clarify how these behaviours were recorded in relation to each other.

We agree that describing the behaviors as mutually exclusive was inaccurate. Our definitions are mutually exclusive because we made them that way. “Cup” is more about the rat’s location and freezing is more about what it is doing. In our scoring procedure cup supersedes freezing because cup behavior is so prevalent during baseline periods. In this data set, freezing with the nose directly above the food cup was uncommon, as freezing in general was present but minimal.

3. Re inter-rater reliability, was there a particular behaviour or category of behaviours that had a lower/higher inter-rater reliability than the others?

This is difficult to determine because there is a wide range in the prevalence of each behavior. For example, in our comparison data set freezing was only observed 5 times, while locomote was detected 69 times and port 100. If you look in Figure 2C, freezing and background are confused more often than freezing and freezing are identically observed. (the freeze x background square is darker than the freeze x freeze square). But all this really means is that there was 4 freeze x backgrounds and 1 freeze x freezes. This speaks to the rarity of freezing more than the reliability of the category. Stretch shows a similar pattern to freezing.

We would say that generally, behaviors based strictly on location (port, cup, jump, rear) show the least confusion. Behaviors that can take place anywhere and play out over frames (stretch, freeze, and locomote) show greater confusion. Much of the confusion is around the start and the stop of the behaviors. This is true for both people and CNNs.

4. Re sex differences: given that the rats were hungry, was body weight included as a covariate that might account for the apparent sex differences?

For Experiment 2, the significant group x sex x session interaction persisted when performing ANCOVA with mean body weight over the 5-day week of behavioral testing was used as a covariate (F_4,92_ = 3.39, *p* = 0.012). Further, the significant group x session interaction (F_4,92_ = 17.51, *p* = 1.02 x 10^-10^) persisted, as did the group x trial x session interaction (F_12,276_ = 2.24, *p* = 0.010).

While performing ANCOVA for Experiment 2, we revisited the Experiment 1 suppression ratio data. We were wondering if body weight was masking significant group x session interactions. It was not. ANCOVA found no significant group x session interaction (F_4,92_ = 0.47, *p* = 1.02 x 10^-10^), and no group x trial x session interaction (F_12,276_ = 2.24, *p* = 0.010).